biomaterials

foam, frog, drug delivery, drug release, antibiotics

**Authors for correspondence:**
Dimitrios A. Lamprou
e-mail: d.lamprou@qub.ac.uk
Paul A. Hoskisson
e-mail: Paul.hoskisson@strath.ac.uk

# Frog nest foams exhibit pharmaceutical foam-like properties

Sarah Brozio[1], Erin M. O'Shaughnessy[2], Stuart Woods[1], Ivan Hall-Barrientos[1], Patricia E. Martin[2], Malcolm W. Kennedy[3], Dimitrios A. Lamprou[4] and Paul A. Hoskisson[1]

[1]Strathclyde Institute of Pharmacy and Biomedical Sciences, University of Strathclyde, 161 Cathedral Street, Glasgow G4 0RE, UK
[2]Department of Biological and Biomedical Sciences, School of Health and Life Sciences, Glasgow Caledonian University, G4 0BA, UK
[3]Institute of Biodiversity Animal Health and Comparative Medicine, Graham Kerr Building, University of Glasgow, Glasgow G12 8QQ, UK
[4]School of Pharmacy, Queen's University Belfast, 97 Lisburn Road, Belfast BT9 7BL, UK

MWK, 0000-0002-0970-5264; PAH, 0000-0003-4332-1640

Foams have frequently been used as systems for the delivery of cosmetic and therapeutic molecules; however, there is high variability in the foamability and long-term stability of synthetic foams. The development of pharmaceutical foams that exhibit desirable foaming properties, delivering appropriate amounts of the active pharmaceutical ingredient (API) and that have excellent biocompatibility is of great interest. The production of stable foams is rare in the natural world; however, certain species of frogs have adopted foam production as a means of providing a protective environment for their eggs and larvae from predators and parasites, to prevent desiccation, to control gaseous exchange, to buffer temperature extremes, and to reduce UV damage. These foams show great stability (up to 10 days in tropical environments) and are highly biocompatible due to the sensitive nature of amphibian skin. This work demonstrates for the first time that nests of the túngara frog (*Engystomops pustulosus*) are stable *ex situ* with useful physiochemical and biocompatible properties and are capable of encapsulating a range of compounds, including antibiotics. These protein foam mixtures share some properties with pharmaceutical foams and may find utility in a range of pharmaceutical applications such as topical drug delivery systems.

# 1. Introduction

Foams have been used as delivery systems or vehicles to deliver cosmetic and therapeutic molecules to normal and injured skin since the 1970s [1–5]. Yet the long-term stability of liquid aqueous foams has been a challenge, with some formulations offering useful foamability properties (e.g. foam expansion time), but poor stability [6]. There has been some progress made through the combination of various foam and surfactant components to create high foamability and long-term foam stability [6–9]. However, the development of biocompatible, liquid foams with high foamability and long-term stability remains a challenge in materials science [6]. A range of foams is already in use for topical treatments, such as Ibuprofen foams (Biatain® Ib) used to relieve a wide range of exuding wounds, urea-containing foams (KerraFoam) to help alleviate the symptoms of psoriasis, and antibiotic foams containing clindamycin and other antimicrobials such as silver sulfacetamide [2,4]. A major advantage of medicated foams is their ability to cover large surface areas, while containing highly concentrated drugs for topical treatment [1]. One major difficulty can be the delivery of an adequate concentration of active pharmaceutical ingredients (APIs) for treatment over a sustained period, therefore often necessitating repeated, regular applications. In the case of open wounds and burns, regular removal of dressings may lead to increased infection risk and damage to healing surfaces, aid the emergence of antimicrobial resistance through the delivery of sub-minimum inhibitory concentrations of antibiotics, ultimately resulting in reduced infection control and wound healing [10]. Liposomes have been proposed for dermatological applications; however, they exhibit major stability issues. There is therefore a need for the development of biomaterials that allow extended times between application combined with high stability and improved biocompatibility; natural foams can provide these benefits.

Anurans (frogs) exhibit an enormous diversity in reproductive strategies and styles [11], and many species of tropical and subtropical frog lay their eggs in stable proteinaceous foams that differ in composition between species. Foam-nesting behaviours, thought to have evolved as a means to avoid aquatic predators, prevent desiccation of eggs, control gaseous exchange, buffer temperature extremes, reduce solar radiation damage and protect eggs from microbial colonization [12]. Stable biological foams and foam-producing surfactants are rare in nature, presumably due to the requirement for high-energy input for their generation, and the potential of surface-active components to negatively affect cell and membrane function [13,14]. Frog nest foams are remarkable for their strong surfactant activity combined with harmlessness to naked eggs and sperm. The leptodactylid frogs of the neotropics are one such anuran lineage that has evolved stable foams as an offspring protection mechanism. The nests of the túngara frog (*Engystomops pustulosus*) are remarkable in structures that act as incubation chambers for eggs [15] (electronic supplementary material, figure S1*a* and *b*), allowing rapid growth and development of embryos, offering a protective environment against predation, while providing temperature regulation and oxygen transfer for optimal growth conditions. These nests are not destroyed by microbes during larval development despite construction within highly microbe-rich water [16]. The foam nest structure is highly stable, remaining assembled for as much as 10 days in the tropical environment [17,18], yet the surfactant activity of the foam does not cause damage to the sperm, eggs or developing embryos [13]. The main surfactant protein within these nests is an 11 kDa protein, Ranaspumin-2 (RSN-2), which does not disrupt biological membranes or cells, but it still provides sufficient surfactant activity in the air–water interface to allow foam formation [19]. The RSN-2 protein appears to form a clamshell-like structure, which can undergo an unfolding conformational change to expose non-polar patches on the protein surface to the air, while highly polar regions remain in contact with the water interface to provide the surfactant activity. RSN-2 has been successfully used in industry as a surfactant in nanoparticle production [20], but there could be great potential for the whole nest foam protein composition to be used in a range of pharmaceutical applications.

Here we show that the unseparated, total protein mixture of *E. pustulosus* nest foam is stable *ex situ* for extended periods with useful physiochemical and biocompatible properties, does not need the addition of oxygen as with other drug delivery systems ([21]; DDS) and may be used to encapsulate a range of hydrophobic and hydrophilic model compounds. These data suggest that anuran-derived protein foams may have broad potential applications in pharmaceutics and as DDSs.

# 2. Results and discussion

## 2.1. Biophysical properties of túngara frog foam preparation

The composition and properties of DDSs can determine the effectiveness of drug release, and while several factors may affect drug release, the thermodynamics driving the passive diffusion process is

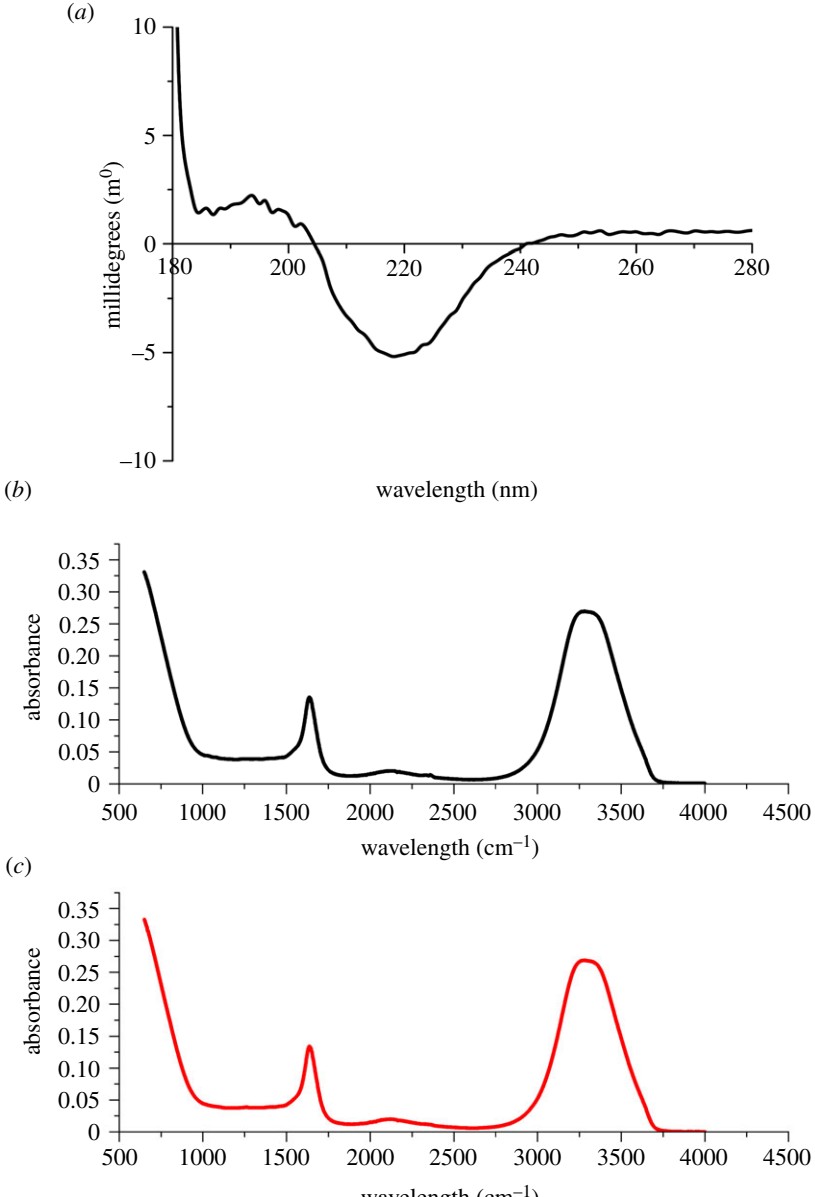

**Figure 1.** Structural characterization of *E. pustulosus* nest foam proteins. (*a*) Circular dichroism of foam fluid using 0.1 mm path length cuvettes containing 1 mg ml$^{-1}$ protein foam fluid solution. (*b*) FTIR foam fluid and (*c*) whole foam. For both CD and FTIR, spectra were corrected for baseline and buffer effects, each measurement was carried out in triplicate, and the mean of the data is presented.

key [22]. Thus, increasing and stabilizing the local concentration of the permeant is the simplest strategy to facilitate bioavailability [3]. The protein composition of *E. pustulosus* foam fluid was analysed by sodium dodecyl sulfate poly-acrylamide gel electrophoresis (SDS-PAGE) (electronic supplementary material, figure S1*c*), confirming previous work that foam from this species contains six major proteins ranging between 10 and 40 kDa in size [18] and that the foam nests used in this study were of typical composition. We observed no variation between collection years or from nests collected at different locations in Trinidad and all foam collected was checked by SDS-PAGE and was of the composition previously detailed in [17–19] and electronic supplementary material, figure S1*c*. Dichroism spectroscopy (CD) spectra of the samples showed a negative maximum at 215 nm and a positive maximum at 194 nm, (figure 1*a*) indicating that, cumulatively, the protein mixture in the foam fluid comprises predominately *β*-sheet structures. Further insight into the secondary structure of the foam mixture was obtained by Fourier transform infrared (FTIR), which also exhibited spectra consistent with the overall dominance of *β*-sheet structures (figure 1*b*; [23]). For both foam solutions,

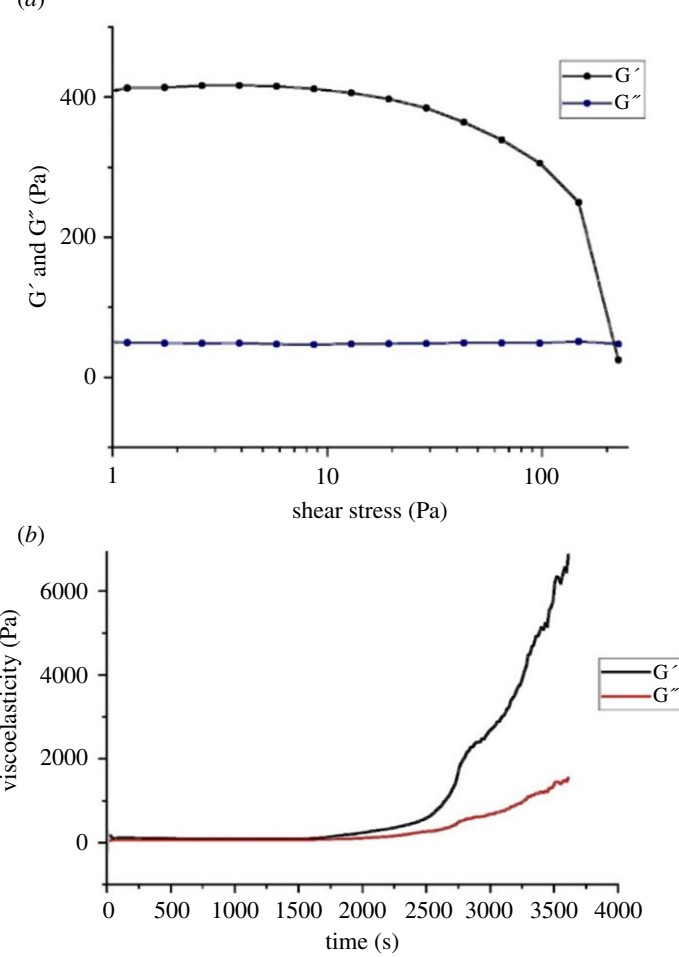

**Figure 2.** Viscoelastic properties of E. pustulosus nest foam. (a) Time sweep rheology data for foam, showing both elastic (G′) and viscous (G″) moduli. Stress was set at 100 Pa and carried out over 1 h. (b) Oscillation sweep rheology data for E. pustulosus foam, showing both elastic (G′) and viscous (G″) moduli. Shear stress was increased from 1 to 200 Pa. Each measurement was taken in triplicate at 20°C.

transitions were observed at approximately 1680 cm$^{-1}$, which are frequencies characteristic of Amide I band, signifying C=O bond stretches typically engaged in β-sheet bonded network structures. These data support previous observations that the average secondary structure content of the proteins is predominantly β-sheet [18] and indicates that the centrifugation steps in the preparation of the foam for these experiments does not alter the overall structure and composition of the foam from the wild E. pustulosus nests.

To investigate the viscoelastic properties of the foam, oscillation sweep experiments were employed by rheology. The whole foam fluid was found to tolerate up to 100 Pa of shear stress force before it reaches a breaking point, at which the elastic modulus of the foam decreases and foam structure and stability is lost (figure 2a). Time sweep experiments indicated that up to 1500 s the foam preparation moduli are unchanged by stress and frequency. After 1500 s, both elastic and viscosity moduli increase, demonstrating that water is being lost from the foam (figure 2b). It has been suggested that stress increases the chances of water loss from the foam followed by coarsening, which leads to an increase in viscoelasticity [24]. The E. pustulosus whole foam is able to withstand shear stress and pressure before breaking down, demonstrating the long-lasting stability that may be observed in nature. Pharmaceutical foams are typically required to remain stable in order to be properly manipulated while being applied, but have low shear, allowing them to break down shortly thereafter [25,26]. The foam derived from E. pustulosus has exhibited long-term stability in harsh tropical environments (e.g. heat, high-level exposure to ultraviolet light and physical disruption) and behaves differently from typical pharmaceutical foams [2]. The E. pustulosus foam is stable enough to be manipulated and able to withstand shear forces, suggesting potential for the delivery of drugs over prolonged periods.

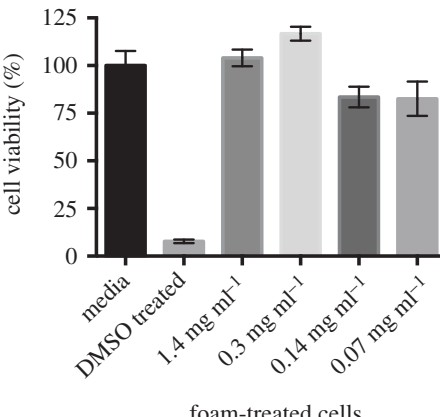

**Figure 3.** Biocompatibility of *E. pustulosus* nest foam with human epithelial cells. MTT assay of HaCaT cell percentage viability following exposure to a range of fluid foam concentrations over 24 h at 37°C. Each treatment was performed in triplicate, and media alone was used as normal viability control (100%), and cells were treated with DMSO for non-viable control. Treatments were a dilution of fluid foam protein concentrations −1.4, 0.3, 0.14 and 0.07 mg ml$^{-1}$, respectively. Error bars represent the standard deviation of the data.

## 2.2. Frog foam biocompatibility with human epithelial cells

To investigate the susceptibility of mammalian cells to any possible toxic effects of the foam, cells were cultured in the presence of foam fluid, and potential toxicity was assayed using an MTT-based cell viability assay. Exposing HaCaT keratinocyte cells to a range of *E. pustulosus* foam concentrations did not affect the overall cell viability and multiplication of the cells (figure 3). The higher foam fluid concentrations to which the cells were exposed are representative of foam concentration present in *E. pustulosus* nests (1–2 mg ml$^{-1}$ protein [17]). Cells exposed to the foam behave in the same manner as untreated control cells, demonstrating that the foam proteins from *E. pustulosus* are non-toxic to epithelial cells and are therefore unlikely to cause damage to the skin or underlying tissues if used as a topical DDS. The foam and its protein components are already known to be harmless to human erythrocytes [27]. This high degree of biocompatibility is consistent with the foam and its precursor components being harmless to naked amphibian sperm, eggs and oviduct surfaces of the frogs [27].

## 2.3. *In vitro* release of drug-loaded frog foam

A single foam cell is defined as a bubble of gas enclosed in a liquid film that can be polyhedral or circular, heterogeneous or homogeneous, and usually ranges between 0.1 and 3 mm in diameter [25]. The cell structure of *E. pustulosus* nest foam was evaluated microscopically and the foam cell sizes measured (figure 4*a*). The foam cells in all samples were found to be a heterogeneous mixture of uneven, spherical and polyhedral cells with a Feret diameter ranging from 10 to 800 µm, falling in the normal range of foam cell size of foams that have been used previously in pharmaceutical applications [25].

The relative density of the foam was found to be 0.25 g of protein ml$^{-1}$. Atomic force microscopy (AFM) PeakForce analysis of the fluid foam and gel foam indicated consistency of the individual adhesion force ($F_{ad}$) measurements in each foam form (figure 4*b*), indicating that the foam surface forces are homogeneous across the surface of each form. Moreover, in the case of the fluid foam, the $F_{ad}$ is higher and the AFM images show the formation of approximately 200 nm droplets as the foam was dried on to mica surfaces. This combination of low density and high structural stability is unusual and suggests anuran foam nest proteins exhibit similar properties to pharmaceutical foams.

To evaluate the drug release behaviour of *E. pustulosus* foam, experiments were performed using a dialysis-based method where two model compounds (one hydrophobic and one hydrophilic), Nile red (NR) and calcein, were encapsulated in the foams. Calcein exhibits a 'burst-type' release profile from the foam, whereas NR is discharged at a linear rate over 7 days (168 h; figure 5*a,b*). These data indicate that the whole foam can absorb and release both hydrophobic (NR) and hydrophilic (calcein) molecules and release these at different rates over a prolonged period, with up to 85% of the loaded dye released up to 7 days (168 h). Longer time periods were not studied, as typically release of APIs from foams occurs on minute to hour timescales [2]; however, 7 days is approaching the lifetime of

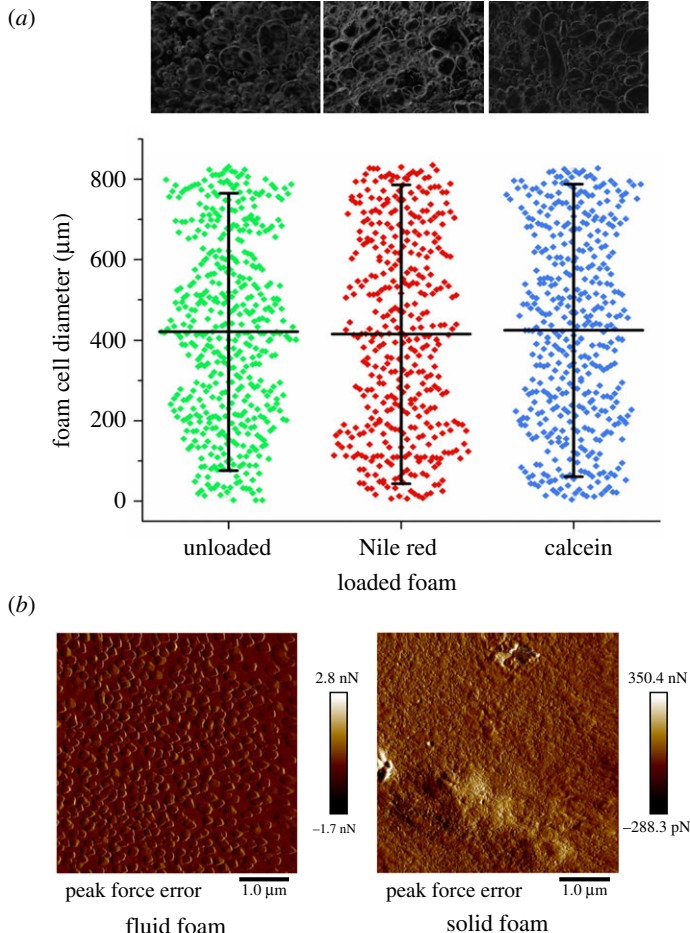

**Figure 4.** Drug loading of *E. pustulosus* nest foam does not alter the structure of the foam. (*a*) Foam cell diameter measurements scatter plot. Feret diameter of each foam cell/bubble was measured using Fiji software. Bars on the scatter encompass 10–90% of the data points, with the central horizontal line representing mean values. Above the scatter graph are representative images of unloaded foam, foam loaded with 1 mg ml$^{-1}$ Nile red (NR) and foam loaded with 1 mg ml$^{-1}$ calcein. All images were taken using freshly defrosted foam. (*b*) AFM PeakForce analysis of foam fluid and gel foam to investigate the consistency of the adhesion force ($F_{ad}$) in the foam.

the foam in the natural environment [15]. Moreover, the loading of the nest foam with NR or calcein did not alter the cell size or shape of the foam (figure 4). The unique release properties associated with the *E. pustulosus* foam may in part be explained by the unique surfactant properties and 'clam shell' structure adopted by at least one member of the protein complex in solution. RSN-2 is an amphiphilic polypeptide which exhibits no obvious hydrophobic patches or structural features that are normally associated with surfactant proteins [14]. RSN-2 is able to adopt an open 'clam-shell' configuration to present the hydrophobic faces of the protein to the interface while maintaining contact of the polar regions of the protein with the aqueous phase. This may enable the *E. pustulosus* foam proteins to modify their structure according to the nature of the drug molecule that is loaded, resulting in the different release profiles for hydrophobic or hydrophilic drug mimics we observed. This property is likely to broaden their potential application as a DDS allowing *E. pustulosus* foam to be loaded with a range of APIs.

To test the ability of the foam to release a clinically relevant drug molecule, the whole foam was loaded in the same manner as for the previous two molecules with the red-pigmented, polyketide antibiotic rifampicin. Polyketide antibiotics have successfully been delivered using foam DDSs, thus rifampicin offers a potentially useful comparison with existing systems [2] and is amenable to spectroscopic analysis. The dialysis method of release showed that rifampicin was released at a steady rate over the first 5 h with a release of around 80% of loaded antibiotic, followed by a second, slower phase of release (figure 5*c*).

To further investigate the release of drug molecules from *E. pustulosus* foam, a novel transwell-based release assay was developed, which emulates delivery of the foam preparation across a more complex protein-coated surface. The foam was loaded into a collagen-coated transwell sat in a 24-well tissue culture

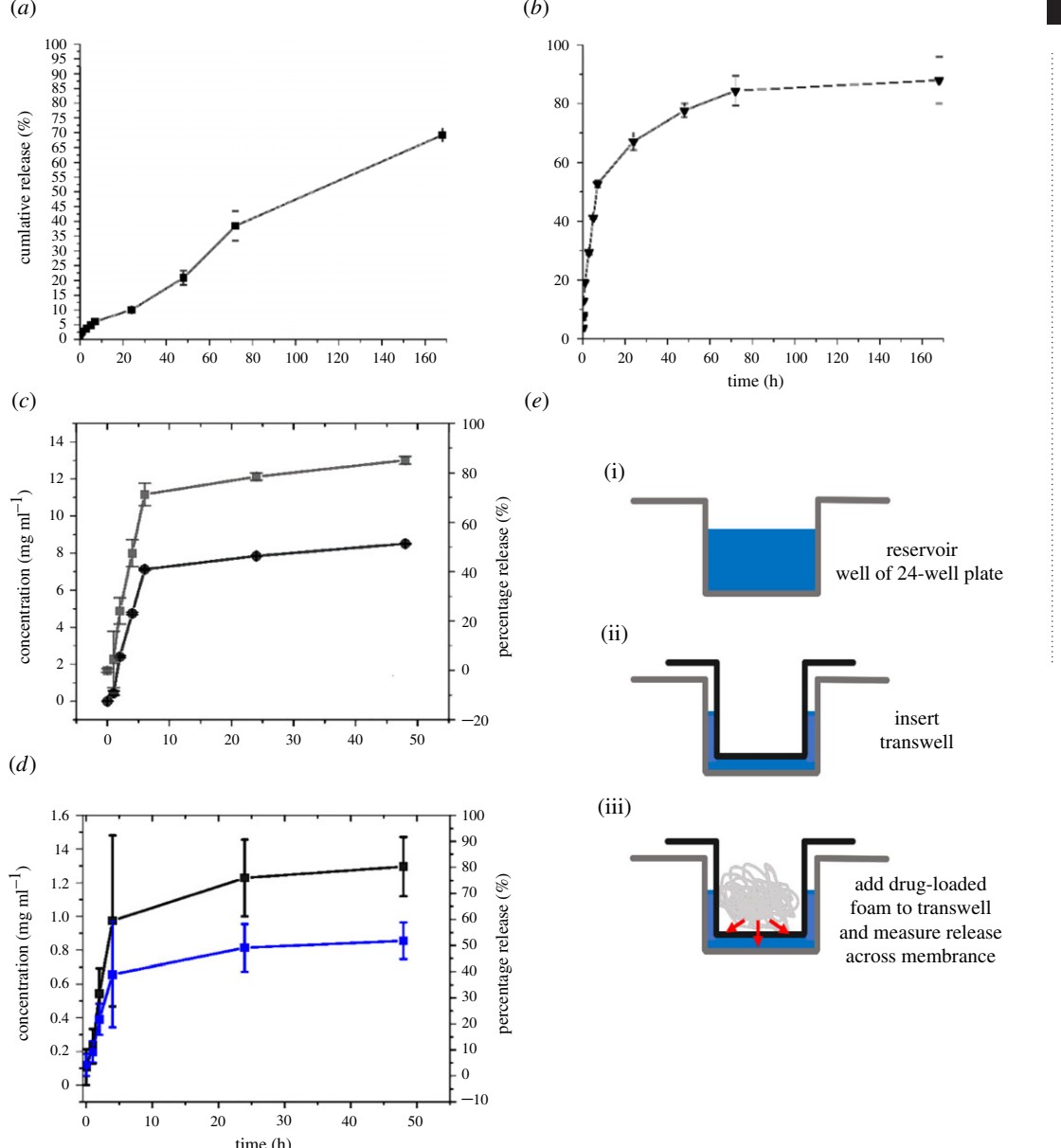

**Figure 5.** *E. pustulosus* nest foam can take up and release model compounds and drug molecules. (*a*) Cumulative release of the dye NR from loaded whole foam over 168 h using the dialysis method. (*b*) Cumulative release of the dye calcein from loaded whole foam over 168 h using the dialysis method. (*c*) Cumulative release of the antibiotic rifampicin from loaded whole foam over 168 h using the dialysis method; circle, concentration (mg ml$^{-1}$); grey square, percentage release of the dye or drug. (*d*) Release of rifampicin using novel transwell method; black square, concentration (mg ml$^{-1}$ of protein); blue square, percentage release of rifampicin. Each point represents the mean of data collected in triplicate and error bars indicate the standard deviation of the data. Sink conditions were satisfied by replacing sample volume with fresh buffer at each sample point. Dye release concentrations were calculated using spectrophotometric standard curves for respective dye or drug. (*e*) Schematic representing the experimental set-up using the novel transwell release assay, with reservoirs of 24-well plate filled with buffer (i), the insertion of the permeable transwell (ii), and the addition of drug-loaded foam to the transwell (iii) to allow drug release across the transwell membrane which can be quantified in the reservoir.

plate containing 1 ml of phosphate-buffered saline (PBS), and release of drug molecule was determined through the assay of the amount of rifampicin that passed from the foam through the transwell membrane. In the transwell assay, around 50% of rifampicin was also released over 48 h (figure 5*d*).

This novel transwell assay for investigating compound release from foams offers a simpler route to assaying drug release from foams and other aqueous-based materials that do not require the manipulation of dialysis tubing. Moreover, the release mimics clinical applications through release from

a single face of the transwell (a schematic of the transwell-based assay is provided in figure 5e) with the advantage that transwells are available in a number of sizes and pore diameters to suit a range of needs.

These data indicate that the frog foam can be loaded with drug molecules and has an extended-release profile when compared with existing pharmaceutical foams, with release over days rather than minutes or hours [28]. The foam from E. pustulosus is stable and the API release is relatively slow with the foam potentially acting as a barrier in the local environment. The E. pustulosus foam also compares well with the release of rifampicin from nanoparticles loaded with antibiotics of the same chemical class, where 80–90% of release occurs, but within an 8 h time window [29].

Many nanoparticle-based systems release their drug load rapidly resulting in ineffective long-term treatment possibilities [30] and have exhibited some toxic properties [31]. The anuran foam-based preparations extend this release period to around 48 h depending on the nature of the compounds used, expanding the possible applications of frog foams. While there have been multiple antibiotic-loaded liposomes [32–34] (AmBisome, Lambin, Doxil) brought to market, little have transferred the applicability to topical conditions, and their efficiency is still lacking. Further, tetracycline (a polyketide antibiotic) loaded nanocomposite hydrogels that have been used for extended-release delivery through the skin, released a maximum of 15% of their antibiotic load [33]. Foams are considered more popular with patients than gels [2], and a stable foam may provide a solution to both of these issues. Previous studies of pharmaceutical foams have investigated the immediate delivery of drugs through to the dermis by exploiting the fast breakdown of foam preparations, but rarely have they been used for long-term drug release [2]. The foam derived from E. pustulosus provides a material with potential due to its intermediate release properties; where API uptake is efficient, it has high stability and the slow release properties enable continuous release over a clinically useful period of time.

# 3. Conclusion

There is an increased interest in DDS to refine the use of antimicrobial drugs currently available on the market, which may enable novel delivery systems to help combat the rise of antimicrobial resistance in the clinic. Anuran foams from reproductive nests may provide a novel area for future investigation in controlled release. Anuran foam nests share properties with pharmaceutical foams, they are highly biocompatible, durable and stable, and have excellent drug release properties. They exhibit a few of the issues associated with fabric-based drug release such as instability, rapid release characteristics or toxicity. These advantages suggest that with further research anuran foams have potential for use as a pharmaceutical foam. To fully meet the demands of modern pharmaceutical manufacture, formulation, sterility, product consistency and for reasons of sustainability, it is envisaged that heterologous production of these proteins will be required for their future exploitation in the clinic. It has been shown that at least one protein is amenable to production in bacterial protein production systems [19]. Further research is required to understand the interaction and properties of the six major foam proteins in combination, and there may be potential to modify the protein mixture using a heterologous expression, for example, simplify the composition of the foam mixture, modify stability with different combinations of component proteins, etc. There is also potential for the use of individual proteins from the nest protein complex as pharmaceutical ingredients given their surfactant properties. It is known that at least one member of the E. pustulosus foam complex (RSN-2) can be whipped to form short-lived (lasting hours rather than days before collapsing) foams that superficially appear similar to natural foam nests [14]. These frog foam nest proteins offer a number of avenues for exploitation in the future, but will require further research to fully exploit their potential.

# 4. Material and methods

## 4.1. Materials

NR, calcein, ethanol 97% (v/v) and PBS tablets (pH 7.2) were all purchased from Sigma, and MTT (3-(4,5-dimethylthiazol-2-yl)-2,5-diphenyltetrazolium) was purchased from Thermofisher.

## 4.2. Collection of Engystomops pustulosus foam

Freshly laid foam nests or adult frogs in amplexus (allowed to lay in captivity, before release) were collected from a number of sites in northern Trinidad in June and July of 2014, 2015 and 2016. Nests

were removed from the surface of the water in which they were produced (an *in situ* nest can be seen in the electronic supplementary material, figure S1*b*), and the eggs were removed manually before being stored at −20°C for transfer to Glasgow, where they were stored at −80°C. Nest foam was pooled and checked by SDS-PAGE to ensure protein integrity (electronic supplementary material, figure S1*c*). Foam was freshly defrosted for all experiments with no further processing being required, as foam maintains its integrity upon freeze/thawing. Soluble foam material (foam fluid) was produced by centrifuging whole foam for 10 min (16 000*g*), providing a solution with approximately 2 mg ml⁻¹ protein. This also yields a supernatant/pellicle residual of semi-solid compressed foam (gel foam) on top of the foam fluid layer.

## 4.3. Microscopy

### 4.3.1. Optical microscopy

The whole foam was defrosted at room temperature before use. All foam images were taken using transmitted light on a Nikon SMZ1500 stereomicroscope with images acquired using a DFK 33UX264 CMOS camera (The Imaging Source Europe GmbH, Germany) using NIS-Elements AR.3.2 software. Fiji software (https://fiji.sc/) from the ImageJ (https://imagej.net) package was used for image analysis.

### 4.3.2. Atomic force microscopy

Samples (5 µl) of foam were deposited onto a freshly cleaved mica surface (1.5 × 1.5 cm; G250-2 Mica sheets 25 × 25 × 0.15 mm; Agar Scientific Ltd, Essex, UK) and left to dry at room temperature for 1 h before imaging. The images were obtained by scanning the mica surface in air under ambient conditions using a Scanning Probe Microscope (MultiMode® 8, Digital Instruments, Santa Barbara, CA, USA; Bruker Nanoscope analysis software v. 1.40), operating using the PeakForce QNM mode. The AFM measurements were obtained using ScanAsyst-air probes, for which the spring constant (0.58 N m⁻¹; nominal 0.4 N m⁻¹) and deflection sensitivity had been calibrated, but not the tip radius (the nominal value used was 2 nm).

## 4.4. Sodium dodecyl sulfate poly-acrylamide gel electrophoresis

Solid and liquid foam samples were electrophoresed on precast NuPAGE 15% poly-acrylamide Bis-Tris gels (Invitrogen) at 120 V using 4 XSDS reducing loading buffer (Invitrogen). Each gel was stained with InstaBlue Coomassie protein stain for approximately 45 min.

## 4.5. Circular dichroism spectroscopy

CD was used to investigate the overall secondary structure content of the proteins. Spectra were acquired using a Chirascan Plus (Applied Photophysics) instrument using a 0.1 mm quartz cuvette (Hellma) at 20°C. All samples (10 mg ml⁻¹ protein) were measured in the far-UV in a wavelength range of 180 to 280 nm, with step size of 1 nm, bandwidth of 1 nm and reading time of 1 s nm⁻¹. Triplicate measurements were taken for each sample run, baseline peak, PBS control and foam sample spectra, with triplicate spectra then averaged. Baseline and PBS traces where subtracted from the sample spectra before secondary structure predictions were made. All data analysis was performed using Global3 software and Excel.

## 4.6. Fourier transform infrared spectroscopy

FTIR spectroscopy was carried out using a Nicolet iS10 Smart iTR spectrophotometer (Thermo Scientific). Solid and liquid foam spectra were recorded in the range of 4000 and 500 cm⁻¹, over 128 scans at a resolution of 4 cm⁻¹ and an interval of 1 cm⁻¹. Background spectra were measured and the foam spectra were corrected accordingly.

## 4.7. Rheology

Rheology measurements were determined using a HAAKE MARS rotational rheometer (Thermo Scientific). Foam samples were subjected to oscillation sweeps and time sweeps. All experiments were

carried out using P20 upper plate and TM20 lower plate. The oscillation sweeps were completed with a 1 mm gap and 0.1 to 200 Pa range. Time sweep experiments were run for 1 h, at 100 Pa and 3 Hz using a 0.5 mm gap. Data points were collected in triplicate and averaged before analysis was carried out.

## 4.8. MTT cell viability assay

HaCaT cells (CLS, Eppelheim, Germany), a model human keratinocyte cell line, were cultured in Dulbecco's modified Eagle's medium (DMEM) containing 4.5 g l$^{-1}$ glucose supplemented with 10% (v/v) fetal bovine serum, 2 mM L-glutamine and 50 units ml$^{-1}$ penicillin/streptomycin (cDMEM; Lonza, Slough, UK). Soluble foam proteins were prepared as above, with the supernatant being passed through the 0.22 µm filter (Millex 33 mm) and subsequently concentrated using an Amicon 10 kDa spin filter. The protein concentration was determined by Bradford assay (BioRad). The HaCaT cells were plated onto 96-well plates (approx. $1 \times 10^3$ cells per well and grown to 80% confluence) and were treated with buffer containing foam proteins (concentrations indicated in the figures) prior to incubation at 37°C for 24 h. After 24 h, the media was removed from the cells and replaced with 50 µl of fresh media and 50 µl of MTT (5 mg ml$^{-1}$) and incubated for 1 h at 37°C. This was followed by replacing the media with 100 µl DMSO and further incubation in the dark at room temperature for 30 min prior to reading the absorbance at 570 nm [35]. Results were expressed as the % cell viability compared with non-treated cells ± s.d. of the data.

## 4.9. *In vitro* release of model compounds

Aliquots (500 mg) of whole foam were loaded with dye by mixing with either 400 µl of NR (hydrophobic; 1 mg ml$^{-1}$ in ethanol) or calcein (hydrophilic; 1 mg ml$^{-1}$ in ethanol). All free liquid containing NR or calcein was encapsulated within the foam following mixing. The mixture was placed in dialysis tubing and sealed before being submerged in 10 ml PBS at 37°C (pH 7; for NR-based release experiments, a 1 : 1 mixture of ethanol and PBS was used). The release experiments were carried out at 37°C, over 168 h. To satisfy the perfect-sink conditions, which allow for the determination of the diffusion parameters, the supernatant was replaced with fresh PBS at 37°C at each time point (indicated in the graphs). The concentration of model compound in each sample was determined spectrophotometrically at 490 nm (calcein) or 590 nm (NR) and the concentration was determined with reference to standard control calibration curves. Experiments were performed in triplicate.

## 4.10. *In vitro* antibiotic release

Two *in vitro* techniques were used to investigate the release of the antibiotic rifampicin.

### 4.10.1. Dialysis

Aliquots (400 mg) of foam were mixed with 400 µl of rifampicin (25 mg ml$^{-1}$) as above. The loaded foam was placed into dialysis tubing, sealed and submerged in 10 ml of PBS. This was incubated at 37°C for 48 h. Samples (1 ml) were taken and fresh media was added to maintain sink conditions. Samples where measured spectrophotometrically at 475 nm [36] against a calibration curve.

### 4.10.2. Transwell

Aliquots of foam (100 mg) were mixed with 100 µl of rifampicin (25 mg ml$^{-1}$). Rifampicin-loaded foam was the placed into a transwell collagen-coated permeable support (0.4 µm; Nunc). Each support was inserted into 24-well plate well containing 600 µl of PBS. The plate was then incubated for 48 h at 37°C. PBS (600 µl) was collected from a well for each time point, and the absorbance was measured at 475 nm, in triplicate.

Ethics. Foam nest collections were approved by the Wildlife Division of Trinidad and Tobago (Special Game Licences 2014–2016). Eggs are removed, undamaged from the foam nests and all tadpoles were returned to the nest collection areas after foam collection. Foam nest proteins were exported under licence (Wildlife Special Export Licence nos. 001741, 001161 and 000646).

Data accessibility. All of the raw data and the electronic supplementary material, data is available on Figshare at https://doi.org/10.6084/m9.figshare.13281416.v2. Data are available on the Dryad Digital Repository at: https://doi.org/10.5061/dryad.hhmgqnkg2 [37].

Authors' contributions. S.B. helped design the study, conducted fieldwork, collected protein samples, collected the data, analysed the data, curated the data and drafted the manuscript; E.M.O., S.W., I.H.-B. and P.E.M. helped during data collection and drafting the manuscript; M.W.K. helped design the study, helped conduct fieldwork and drafting the manuscript; D.A.L. helped conceive and design the study and in drafting the manuscript; P.A.H. conceived and designed the study, conducted fieldwork, collected protein samples and drafted the manuscript. All authors gave final approval for publication and agree to be held accountable for the work performed therein.

Competing interests. All authors declare that they have no conflict of interest in relation to this work.

Funding. The authors would like to acknowledge the Engineering and Physical Science Research Council (EPSRC) via the Doctoral Training Centre (DTC) at the University of Strathclyde for the PhD studentship support to S.B.

Acknowledgements. The authors would like to acknowledge the Engineering and Physical Science Research Council (EPSRC) via the Doctoral Training Centre (DTC) at the University of Strathclyde for the PhD studentship support to S.B. We would also like to thank Prof. Roger Downie, University of Glasgow for his long-term assistance in the field and advice on túngara frogs. We acknowledge the support of the Microbiology Society and the Pauline Fitzpatrick Memorial Travel Fund to S.B. to support fieldwork in Trinidad. E.M.O. was supported by a PhD studentship from the Psoriasis Association (ST3 15). We would also like to thank the Wildlife Section, Forestry Division, of the Government of Trinidad and Tobago for issuing Special Game Licences under the Conservation of Wildlife Act, permitting us to collect *E. pustulosus* nests (Special Game Licences 2014–2016 and Wildlife Special Export Licence nos. 001741, 001161 and 000646).

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
