## [Peer Review File · Royal Society Open Science]

Review History

RSOS-210048.R0 (Original submission)

Review form: Reviewer 1 (Munira M. Momin)

Is the manuscript scientifically sound in its present form?

Yes

Are the interpretations and conclusions justified by the results?

Yes

Is the language acceptable?

Yes

Do you have any ethical concerns with this paper?

No

Have you any concerns about statistical analyses in this paper?

Yes

Recommendation?

Accept with minor revision (please list in comments)

Comments to the Author(s)

The author needs to comment on the source variation/ natural habitat/ effect of climatic condition/ uniformity of the material with variation in place and weather.

Review form: Reviewer 2**Is the manuscript scientifically sound in its present form?**

No

Are the interpretations and conclusions justified by the results?

No

Is the language acceptable?

Yes

Do you have any ethical concerns with this paper?

No

Have you any concerns about statistical analyses in this paper?

No

Recommendation?

Reject

Comments to the Author(s)

Observing nature and using its solutions is always a very good idea. The authors of the article "Frog nest Foam as a drug delivery system" have a very good idea of using natural substances as dosage forms. However, there are many problems in the work with pharmaceutical technology.

1) Foam was collected and hand cleaned of eggs, and what about other debris. Part of the foam was washed off the frog skin - the authors did not specify what dilution they received as a result and whether this solution was combined with the foam from the same frog or stored separately

2) Were the frog's foams mixed or frozen separately? And if they were not combined, were the individual foams compared in some way (the content of proteins responsible for foaming, foam cell size distribution, etc.). If they were combined, how were the charges unified?

3) The authors write that before the tests, the foam samples were freshly defrosted - does this mean that the foam retained its structure in the process of freezing and defrosting, or was it reconstructed in some way? How?

4) How exactly was the foam combined with the antibiotic? - technological processes in the production of the drug form must be strictly defined and must be repeatable

5) What exactly is this form of the drug intended for? It is necessary to determine this in order to determine the purity class of the drug and properly select the set of tests describing the resulting form of the drug. If the foam is to be used on wounds, it must meet the sterility requirements, is the foam - and more specifically the protein responsible for foaming - resistant to sterilization conditions? However, if it is to be applied to the skin, the parameters that should be described must be spreadability and consistency.

These are just a few comments that arise after reading the article, unfortunately, there are many more of them. Although the idea of using foam is really interesting, the authors of the study did not take into account the assumptions of modern pharmaceutical technology. They did not pay enough attention to the fact that the form of the drug requires careful planning of all elements of the experiment so that the obtained products meet the pharmacopeial standards. For this reason, I believe that the work is not suitable for publishing at this stage of research.

Review form: Reviewer 3

Is the manuscript scientifically sound in its present form?

Yes

Are the interpretations and conclusions justified by the results?

Yes

Is the language acceptable?

Yes

Do you have any ethical concerns with this paper?

No

Have you any concerns about statistical analyses in this paper?

No

Recommendation?

Accept with minor revision (please list in comments)

Comments to the Author(s)

In this manuscript the authors introduce the use of foams derived from frogs as a drug delivery system for pharmaceutical applications. This is novel approach that could potentially be further developed. The authors have conducted a load of experimntal work including several analytical techniques. The discussion is well written and in a good agreement with the experimental data. I recommend the article for publication after some minor adjustments:

-In Fig. 3 the authors should present the MTT findings as percentage (%) of the viable cells and not the absorbance values which are not easy to follow.

- What was the drug loading in the foams? This should be presented for the dyes and the drug substance used for the purpose of the work.

- How do the authors explain that the release patterns for the hydrophilic pigments is slower compared to the relative hydrophobic drugs rifampicin.

Decision letter (RSOS-210048.R0)

Dear Dr Hoskisson

The Editors assigned to your paper RSOS-210048 "Frog nest foam as a drug delivery system" have now received comments from reviewers and would like you to revise the paper in accordance with the reviewer comments and any comments from the Editors. Please note this decision does not guarantee eventual acceptance.

Please submit your revised manuscript and required files (see below) no later than 21 days from today's (ie 14-Jun-2021) date. Note: the ScholarOne system will 'lock' if submission of the revision is attempted 21 or more days after the deadline. If you do not think you will be able to meet this deadline please contact the editorial office immediately.

on behalf of Professor Malcolm White (Subject Editor)
openscience@royalsociety.org

Associate Editor Comments to Author:

While two of the reviewers have offered positive feedback on your work, given the concerns raised by reviewer 2, we would like you to revise the paper, taking particular care to address those concerns. If you are able to revise the paper and provide a rebuttal that satisfies the concerns of this reviewer, we would be glad to reconsider the paper for publication, but if the referee remains of the view that the paper cannot be taken forward, we regret we may not be able to consider the manuscript further. We'll look forward to seeing how you engage with the queries and comments.

Reviewer comments to Author:

Reviewer: 1

Comments to the Author(s)

The author needs to comment on the source variation/ natural habitat/ effect of climatic condition/ uniformity of the material with variation in place and weather.

Reviewer: 2

Comments to the Author(s)

Observing nature and using its solutions is always a very good idea. The authors of the article "Frog nest Foam as a drug delivery system" have a very good idea of using natural substances as dosage forms. However, there are many problems in the work with pharmaceutical technology.

1) Foam was collected and hand cleaned of eggs, and what about other debris. Part of the foam was washed off the frog skin - the authors did not specify what dilution they received as a result and whether this solution was combined with the foam from the same frog or stored separately

2) Were the frog's foams mixed or frozen separately? And if they were not combined, were the individual foams compared in some way (the content of proteins responsible for foaming, foam cell size distribution, etc.). If they were combined, how were the charges unified?

3) The authors write that before the tests, the foam samples were freshly defrosted - does this mean that the foam retained its structure in the process of freezing and defrosting, or was it reconstructed in some way? How?

4) How exactly was the foam combined with the antibiotic? - technological processes in the production of the drug form must be strictly defined and must be repeatable

5) What exactly is this form of the drug intended for? It is necessary to determine this in order to determine the purity class of the drug and properly select the set of tests describing the resulting form of the drug. If the foam is to be used on wounds, it must meet the sterility requirements, is the foam - and more specifically the protein responsible for foaming - resistant to sterilization conditions? However, if it is to be applied to the skin, the parameters that should be described must be spreadability and consistency.

These are just a few comments that arise after reading the article, unfortunately, there are many more of them. Although the idea of using foam is really interesting, the authors of the study did not take into account the assumptions of modern pharmaceutical technology. They did not pay enough attention to the fact that the form of the drug requires careful planning of all elements of the experiment so that the obtained products meet the pharmacopeial standards. For this reason, I believe that the work is not suitable for publishing at this stage of research.

Reviewer: 3

Comments to the Author(s)

In this manuscript the authors introduce the use of foams derived from frogs as a drug delivery system for pharmaceutical applications. This is novel approach that could potentially be further developed. The authors have conducted a lot of experimental work including several analytical techniques. The discussion is well written and in a good agreement with the experimental data. I recommend the article for publication after some minor adjustments:

-In Fig. 3 the authors should present the MTT findings as percentage (%) of the viable cells and not the absorbance values which are not easy to follow.

- What was the drug loading in the foams? This should be presented for the dyes and the drug substance used for the purpose of the work.
- How do the authors explain that the release patterns for the hydrophilic pigments is slower compared to the relative hydrophobic drugs rifampicin.

===PREPARING YOUR MANUSCRIPT===

===PREPARING YOUR REVISION IN SCHOLARONE===

<https://royalsociety.org/journals/authors/author-guidelines/#supplementary-material> to include a suitable title and informative caption. An example of appropriate titling and captioning may be found at https://figshare.com/articles/Table_S2_from_Is_there_a_trade-off_between_peak_performance_and_performance_breadth_across_temperatures_for_aerobic_sc_ope_in_teleost_fishes_/3843624.

Author's Response to Decision Letter for (RSOS-210048.R0)

See Appendix A.

RSOS-210048.R1 (Revision)

Review form: Reviewer 2

Is the manuscript scientifically sound in its present form?

No

Are the interpretations and conclusions justified by the results?

No

Is the language acceptable?

Yes

Do you have any ethical concerns with this paper?

No

Have you any concerns about statistical analyses in this paper?

No

Recommendation?

Reject

Comments to the Author(s)

Dear Editor,

I regret to say that the authors did not read my review correctly. The manuscript in its current, already "improved" form, does not represent state of the art work concerning rational development of medicines. It does not contain elements related to pharmaceutical technology. As mentioned in an earlier review: " Authors did not pay enough attention to the fact that the development of foams as unique drug delivery devices requires careful planning of all aspects including the selection of excipients, characterization techniques and the application . There are still many elements that are irrational, e.g. why should the foam be stable for 168 hours? Why was rifampicin used as the model drug? It is an antibiotic reserved for the treatment of tuberculosis (not applied to the skin)? e.t.c.

Considering the above mentioned major drawbacks the manuscript in the present form as well as the generated data are not suitable for publication in the field of biopharmaceutics and pharmaceutical technology

Decision letter (RSOS-210048.R1)

Dear Dr Hoskisson

The Editors assigned to your paper RSOS-210048.R1 "Frog nest foam as a drug delivery system" have now received comments from reviewers and would like you to revise the paper in accordance with the reviewer comments and any comments from the Editors. Please note this decision does not guarantee eventual acceptance.

We do not generally allow multiple rounds of revision so we urge you to make every effort to fully address all of the comments at this stage. If deemed necessary by the Editors, your

manuscript will be sent back to one or more of the original reviewers for assessment. If the original reviewers are not available, we may invite new reviewers.

Please submit your revised manuscript and required files (see below) no later than 21 days from today's (ie 30-Jul-2021) date. Note: the ScholarOne system will 'lock' if submission of the revision is attempted 21 or more days after the deadline. If you do not think you will be able to meet this deadline please contact the editorial office immediately.

on behalf of Prof Malcolm White (Subject Editor)
openscience@royalsociety.org

Editor Comments to Author:

It is unusual to offer an additional opportunity to revise a paper in the journal, but the opportunity to revise has a number of caveats. Firstly, you should do your best to amend the paper to highlight the concerns raised by this reviewer, but the paper is probably publishable as long as it doesn't over claim. Given this, the second caveat is that you should change the title to "potential drug delivery system" or similar. Assuming you take these caveats into account, the paper should not need to go back to the critical reviewer, but will receive final assessment by the editors.

Reviewer comments to Author:

Reviewer: 2

Comments to the Author(s)

Dear Editor,

I regret to say that the authors did not read my review correctly. The manuscript in its current, already "improved" form, does not represent state of the art work concerning rational development of medicines. It does not contain elements related to pharmaceutical technology. As mentioned in an earlier review: "Authors did not pay enough attention to the fact that the development of foams as unique drug delivery devices requires careful planning of all aspects including the selection of excipients, characterization techniques and the application. There are still many elements that are irrational, e.g. why should the foam be stable for 168 hours? Why was rifampicin used as the model drug? It is an antibiotic reserved for the treatment of tuberculosis (not applied to the skin)? e.t.c.

Considering the above mentioned major drawbacks the manuscript in the present form as well as the generated data are not suitable for publication in the field of biopharmaceutics and pharmaceutical technology

===PREPARING YOUR MANUSCRIPT===

===PREPARING YOUR REVISION IN SCHOLARONE===

<https://royalsociety.org/journals/authors/author-guidelines/#supplementary-material> to include a suitable title and informative caption. An example of appropriate titling and captioning may be found at https://figshare.com/articles/Table_S2_from_Is_there_a_trade-off_between_peak_performance_and_performance_breadth_across_temperatures_for_aerobic_sc_ope_in_teleost_fishes_/3843624.

Author's Response to Decision Letter for (RSOS-210048.R1)

See Appendix B.

Decision letter (RSOS-210048.R2)

Dear Dr Hoskisson,

It is a pleasure to accept your manuscript entitled "Frog nest foams exhibit pharmaceutical foam-like properties" in its current form for publication in Royal Society Open Science. The comments of the reviewer(s) who reviewed your manuscript are included at the foot of this letter.

You can expect to receive a proof of your article in the near future. Please contact the editorial office (openscience@royalsociety.org) and the production office (openscience_proofs@royalsociety.org) to let us know if you are likely to be away from e-mail contact -- if you are going to be away, please nominate a co-author (if available) to manage the proofing process, and ensure they are copied into your email to the journal. We note that the email address for one of your co-authors is currently marked as 'invalid'. Please could you kindly reply to this email with an updated email address for: erin.oshaughnessy@merckgroup.com

on behalf of Professor Malcolm White (Subject Editor)
openscience@royalsociety.org

Appendix A

Authors Response to comments

Associate Editor Comments to Author:

While two of the reviewers have offered positive feedback on your work, given the concerns raised by reviewer 2, we would like you to revise the paper, taking particular care to address those concerns. If you are able to revise the paper and provide a rebuttal that satisfies the concerns of this reviewer, we would be glad to reconsider the paper for publication, but if the referee remains of the view that the paper cannot be taken forward, we regret we may not be able to consider the manuscript further. We'll look forward to seeing how you engage with the queries and comments.

Response: We would like to thank the Referees and Associate Editor for taking the time to examine our manuscript and we hope that we have addressed all of the comments satisfactorily in the revised manuscript, as detailed below.

Reviewer comments to Author:

Reviewer: 1

Comments to the Author(s)

The author needs to comment on the source variation/ natural habitat/ effect of climatic condition/ uniformity of the material with variation in place and weather.

Response: We would like to thank the reviewer for their comments and we have added additional lines to the Results & Discussion and materials & methods to clarify this point (lines 106-109 and 324-325). But essentially, we observed no variation in foam composition between frogs, locations around Trinidad or when compared with samples collected in previous studies (see references Fleming *et al.*, 2009; *Proceedings of the Royal Society B: Biological Sciences* **276**. (doi:10.1098/rspb.2008.1939); Cooper *et al.*, 2005 *Biophys J* **88**, 2114–2125. (doi:10.1529/biophysj.104.046268); Mackenzie *et al.*, 2009 *Biophys J* **96**. (doi:10.1016/j.bpj.2009.03.044). We have also added additional text to the conclusions to indicate that we see clinical use of this foam being from heterologously produced protein (lines 298-311)

Reviewer: 2

Comments to the Author(s)

Observing nature and using its solutions is always a very good idea. The authors of the article "Frog nest Foam as a drug delivery system" have a very good idea of using natural substances as dosage forms. However, there are many problems in the work with pharmaceutical technology.

Response: We thank the referee for recognising the potential for such bioinspired materials and hopefully we have allayed their concerns with our revisions to the manuscript. We believe that there is great potential for the frog foams as a pharmaceutical agent, but also agree with the referee that there are a number of manufacturing and formulation hurdles to overcome before we see them used in the clinic. We also envisage that individual foam proteins would be manufactured as recombinant proteins rather than collected from wild frogs. This is an area that we are currently pursuing and is beyond the scope of the current manuscript. We have added additional text to the discussion to clarify this to readers (lines 227-237) and amended the abstract, introduction and conclusions to help clarify for readers (lines 33-34, 93-94, 298-311).

1) Foam was collected and hand cleaned of eggs, and what about other debris. Part of the foam was washed off the frog skin - the authors did not specify what dilution they received as a result and whether this solution was combined with the foam from the same frog or stored separately

Response: We collected the foam either from frogs that laid the eggs in clean water in the laboratory, or from fresh nests laid in the wild. There is no washing from the frog's skin. Any parts of the nest that contained debris were avoided to ensure that the foam remained as uncontaminated as possible. The foam is not diluted during the collection process. Foam from a number of nests is pooled. Previous work (Fleming *et al.*, 2009; *Proceedings of the Royal Society B: Biological Sciences* **276**. (doi:10.1098/rspb.2008.1939); Cooper *et al.*, 2005 *Biophys J* **88**, 2114–2125. (doi:10.1529/biophysj.104.046268); Mackenzie *et al.*, 2009 *Biophys J* **96**. (doi:10.1016/j.bpj.2009.03.044) and our studies showed that the foam composition (via SDS-PAGE; Supp. Data Fig 1) was consistent across frog populations on Trinidad and over a number of years (>10 years). We have added additional text to the Materials and Methods section to clarify this point (Lines 106-109, 324-325).

2) Were the frog's foams mixed or frozen separately? And if they were not combined, were the individual foams compared in some way (the content of proteins responsible for foaming, foam cell size distribution, etc.). If they were combined, how were the charges unified?

Response: We pooled foam samples from a number of frogs. It is known from previous work by Professor Kennedy (a co-author on this work; Fleming *et al.*, 2009; *Proceedings of the Royal Society B: Biological Sciences* **276**. (doi:10.1098/rspb.2008.1939); Cooper *et al.*, 2005 *Biophys J* **88**, 2114–2125. (doi:10.1529/biophysj.104.046268); Mackenzie *et al.*, 2009 *Biophys J* **96**. (doi:10.1016/j.bpj.2009.03.044) that the foam between frogs and over >10 year period are consistent in composition (see points above). Experiments were performed using several frog foam samples, batches and experiments were conducted over several years showed very similar results. We have clarified this in lines 106-109 & lines 324-325.

3) The authors write that before the tests, the foam samples were freshly defrosted - does this mean that the foam retained its structure in the process of freezing and defrosting, or was it reconstructed in some way? How?

Response: The foam retains its structure following freezing and defrosting. There was no need to reconstitute the foam. We have updated the text to indicate this in the text (lines 326-327).

4) How exactly was the foam combined with the antibiotic? - technological processes in the production of the drug form must be strictly defined and must be repeatable

Response: The encapsulation efficacy was ~100%, as the drug-mimic or antibiotic used (400 µl) was directly encapsulated when mixed with the foam at 1 mg/ml concentration. All free liquid was encapsulated by the foam following mixing. We have added additional text to clarify this in the Materials & Methods section (Lines 388-389).

5) What exactly is this form of the drug intended for? It is necessary to determine this in order to determine the purity class of the drug and properly select the set of tests describing the resulting form of the drug. If the foam is to be used on wounds, it must meet the sterility requirements, is the foam - and more specifically the protein responsible for foaming - resistant to sterilization conditions? However, if it is to be applied to the skin, the parameters that should be described must be spreadability and consistency.

Response: We used the antibiotic as a model to indicate that the foam can be loaded with a 'real' drug, rather than just the 'dye drug-mimics'. We are proposing that the foam could be used for topical application and delivery of drugs but certainly not limited to this. We would expect that any foam used clinically would be produced as a heterologous protein, rather than collected from the wild, to ensure that it met the standards required for pharmaceutical application. We envisage that the whole foam complex could find utility, but also it may be that individual foam proteins, with their surfactant properties may also find utility in creams and cosmetics potentially. It was not our aim to ascribe a definitive use or purpose to the foam proteins, simply to demonstrate their broad utility. We have added additional text to the conclusions to indicate the potential of the foam proteins (lines 298-311).

These are just a few comments that arise after reading the article, unfortunately, there are many more of them. Although the idea of using foam is really interesting, the authors of the study did not take into account the assumptions of modern pharmaceutical technology. They did not pay enough attention to the fact that the form of the drug requires careful planning of all elements of the experiment so that the obtained products meet the pharmacopeial standards. For this reason, I believe that the work is not suitable for publishing at this stage of research.

Response: We agree with the referee, there is a large number of considerations in terms of the use of the foam in a clinical setting and how this may be manufactured within the standards of the Pharmacopoeia. We did not intend the manuscript to be a preclinical study. We believe that we are at the beginning of understanding how these foams may be exploited and attempted to show the potential of these foams. We have added some additional text to the discussion in terms of how we think these foams may translate in to a pharmaceutical product following on from our findings (lines 294, 298-311).

Reviewer: 3

Comments to the Author(s)

In this manuscript the authors introduce the use of foams derived from frogs as a drug delivery system for pharmaceutical applications. This is novel approach that could potentially be further developed. The authors have conducted a lot of experimntal work including several analytical techniques. The discussion is well written and in a good agreement with the experimental data. I recommend the article for publication after some minor adjustments:

Response: We thank the referee for their positive comments on our work, recognising the novelty of our work and hopefully have addressed their concerns below.

-In Fig. 3 the authors should present the MTT findings as percentage (%) of the viable cells and not the absorbance values which are not easy to follow.

Response: We have replotted the data for the MTT as a percentage of the viable cells as suggested for Fig. 3 and updated the figure and legend in the manuscript. The raw data file submitted with the manuscript already had these data in and plotted so this does not need to be updated.

- What was the drug loading in the foams? This should be presented for the dyes and the drug substance used for the purpose of the work.

Response: The encapsulation efficacy was ~100%, as the drug-mimic or antibiotic used (400 µl) was directly encapsulated/"mixed" with the foam at 1 mg/ml concentration. All free liquid was encapsulated by the foam following mixing. We have added additional text to clarify this in the Materials & Methods section (Lines 383-384, 402-403).

- How do the authors explain that the release patterns for the hydrophilic pigments is slower compared to the relative hydrophobic drugs rifampicin.

Response: We believe that the differences in release patterns are related to the this is due to the interaction between the drug molecule and the novel surfactant properties of these foam proteins. We have added some text (lines 224-233) to details this in light of previous structural work that was undertaken on one of the foam proteins. We believe that this is a further example of how unique this foam approach is for the delivery of both molecules.

Appendix B

STRATHCLYDE INSTITUTE OF PHARMACY & BIOMEDICAL SCIENCES

28th June 2021

RE: Submission of article entitled 'Frog nest foam as a drug delivery system'

Dear Professor White,

Thank you and the three referees for considering our manuscript and on behalf of all authors please find attached our revised article '**Frog nest foam as a drug delivery system**'.

We have taken on board all of the referee's comments and amended our manuscript accordingly. In particular, in response to the concerns of referee #2, we have added additional discussion around how we think these proteins could be used in the pharmaceutical industry, with reference to the use of heterologously produced proteins to meet the pharmacopeia standards. It was not our intention for this work to be a pre-clinical study, merely to demonstrate that these proteins have great potential utility and to demonstrate one such use. We have also added the methodological details requested by the referees relating to consistency of the foam and some discussion around how the structural properties of contribute to the release pattern differences we observed.

All authors have agreed to resubmission of the manuscript.

I look forward to hearing from you in due course,

With kind regards and many thanks,

Professor Paul A Hoskisson FRSE FRSB
Royal Academy of Engineering Research Chair in Engineering Biology and Chair of Molecular Microbiology
University of Strathclyde,
email: paul.hoskisson@strath.ac.uk

The place of useful learning

The University of Strathclyde is a charitable body, registered in Scotland, number SC015263

Authors Response to comments

Associate Editor Comments to Author:

While two of the reviewers have offered positive feedback on your work, given the concerns raised by reviewer 2, we would like you to revise the paper, taking particular care to address those concerns. If you are able to revise the paper and provide a rebuttal that satisfies the concerns of this reviewer, we would be glad to reconsider the paper for publication, but if the referee remains of the view that the paper cannot be taken forward, we regret we may not be able to consider the manuscript further. We'll look forward to seeing how you engage with the queries and comments.

Response: We would like to thank the Referees and Associate Editor for taking the time to examine our manuscript and we hope that we have addressed all of the comments satisfactorily in the revised manuscript, as detailed below.

Reviewer comments to Author:

Reviewer: 1

Comments to the Author(s)

The author needs to comment on the source variation/ natural habitat/ effect of climatic condition/ uniformity of the material with variation in place and weather.

Response: We would like to thank the reviewer for their comments and we have added an additional lines to the Results & Discussion and materials & methods to clarify this point (lines 106-109 and 324-325). But essentially, we observed no variation in foam composition between frogs, locations around Trinidad or when compared with samples collected in previous studies (see references Fleming *et al.*, 2009; *Proceedings of the Royal Society B: Biological Sciences* **276**. (doi:10.1098/rspb.2008.1939); Cooper *et al.*, 2005 *Biophys J* **88**, 2114–2125. (doi:10.1529/biophysj.104.046268); Mackenzie *et al.*, 2009 *Biophys J* **96**. (doi:10.1016/j.bpj.2009.03.044). We have also added additional text to the conclusions to indicate that we see clinical use of this foam being from heterologously produced protein (lines 298-311)

Reviewer: 2

Comments to the Author(s)

Observing nature and using its solutions is always a very good idea. The authors of the article "Frog nest Foam as a drug delivery system" have a very good idea of using natural substances as dosage forms. However, there are many problems in the work with pharmaceutical technology.

Response: We thank the referee for recognising the potential for such bioinspired materials and hopefully we have allayed their concerns with our revisions to the manuscript. We believe that there is

The place of useful learning

The University of Strathclyde is a charitable body, registered in Scotland, number SC015263

great potential for the frog foams as a pharmaceutical agent, but also agree with the referee that there are a number of manufacturing and formulation hurdles to overcome before we see them used in the clinic. We also envisage that individual foam proteins would be manufactured as recombinant proteins rather than collected from wild frogs. This is an area that we are currently pursuing and is beyond the scope of the current manuscript. We have added additional text to the discussion to clarify this to readers (lines 227-237) and amended the abstract, introduction and conclusions to help clarify for readers (lines 33-34, 93-94, 298-311).

1) Foam was collected and hand cleaned of eggs, and what about other debris. Part of the foam was washed off the frog skin - the authors did not specify what dilution they received as a result and whether this solution was combined with the foam from the same frog or stored separately

Response: We collected the foam either from frogs that laid the eggs in clean water in the laboratory, or from fresh nests laid in the wild. There is no washing from the frog's skin. Any parts of the nest that contained debris were avoided to ensure that the foam remained as uncontaminated as possible. The foam is not diluted during the collection process. Foam from a number of nests is pooled. Previous work (Fleming *et al.*, 2009; *Proceedings of the Royal Society B: Biological Sciences* **276**. (doi:10.1098/rspb.2008.1939); Cooper *et al.*, 2005 *Biophys J* **88**, 2114–2125. (doi:10.1529/biophysj.104.046268); Mackenzie *et al.*, 2009 *Biophys J* **96**. (doi:10.1016/j.bpj.2009.03.044) and our studies showed that the foam composition (via SDS-PAGE; Supp. Data Fig 1) was consistent across frog populations on Trinidad and over a number of years (>10 years). We have added additional text to the Materials and Methods section to clarify this point (Lines 106-109, 324-325).

2) Were the frog's foams mixed or frozen separately? And if they were not combined, were the individual foams compared in some way (the content of proteins responsible for foaming, foam cell size distribution, etc.). If they were combined, how were the charges unified?

Response: We pooled foam samples from a number of frogs. It is known from previous work by Professor Kennedy (a co-author on this work; Fleming *et al.*, 2009; *Proceedings of the Royal Society B: Biological Sciences* **276**. (doi:10.1098/rspb.2008.1939); Cooper *et al.*, 2005 *Biophys J* **88**, 2114–2125. (doi:10.1529/biophysj.104.046268); Mackenzie *et al.*, 2009 *Biophys J* **96**. (doi:10.1016/j.bpj.2009.03.044) that the foam between frogs and over >10 year period are consistent in composition (see points above). Experiments were performed using several frog foam samples, batches and experiments were conducted over several years showed very similar results. We have clarified this in lines 106-109 & lines 324-325.

3) The authors write that before the tests, the foam samples were freshly defrosted - does this mean that the foam retained its structure in the process of freezing and defrosting, or was it reconstructed in some way? How?

Response: The foam retains its structure following freezing and defrosting. There was no need to reconstitute the foam. We have updated the text to indicate this in the text (lines 326-327).

4) How exactly was the foam combined with the antibiotic? - technological processes in the production of the drug form must be strictly defined and must be repeatable

Response: The encapsulation efficacy was ~100%, as the drug-mimic or antibiotic used (400 μ l) was directly encapsulated when mixed with the foam at 1 mg/ml concentration. All free liquid was encapsulated by the foam following mixing. We have added additional text to clarify this in the Materials & Methods section (Lines 388-389).

5) What exactly is this form of the drug intended for? It is necessary to determine this in order to determine the purity class of the drug and properly select the set of tests describing the resulting form of the drug. If the foam is to be used on wounds, it must meet the sterility requirements, is the foam - and more specifically the protein responsible for foaming - resistant to sterilization conditions? However, if it is to be applied to the skin, the parameters that should be described must be spreadability and consistency.

Response: We used the antibiotic as a model to indicate that the foam can be loaded with a 'real' drug, rather than just the 'dye drug-mimics'. We are proposing that the foam could be used for topical application and delivery of drugs but certainly not limited to this. We would expect that any foam used clinically would be produced as a heterologous protein, rather than collected from the wild, to ensure that it met the standards required for pharmaceutical application. We envisage that the whole foam complex could find utility, but also it may be that individual foam proteins, with their surfactant properties may also find utility in creams and cosmetics potentially. It was not our aim to ascribe a definitive use or purpose to the foam proteins, simply to demonstrate their broad utility. We have added additional text to the conclusions to indicate the potential of the foam proteins (lines 298-311).

These are just a few comments that arise after reading the article, unfortunately, there are many more of them. Although the idea of using foam is really interesting, the authors of the study did not take into account the assumptions of modern pharmaceutical technology. They did not pay enough attention to the fact that the form of the drug requires careful planning of all elements of the experiment so that the obtained products meet the pharmacopeial standards. For this reason, I believe that the work is not suitable for publishing at this stage of research.

Response: We agree with the referee, there is a large number of considerations in terms of the use of the foam in a clinical setting and how this may be manufactured within the standards of the Pharmacopoeia. We did not intend the manuscript to be a preclinical study. We believe that we are at the beginning of understanding how these foams may be exploited and attempted to show the potential of

The place of useful learning

these foams. We have added some additional text to the discussion in terms of how we think these foams may translate in to a pharmaceutical product following on from our findings (lines 294, 298-311).

Reviewer: 3

Comments to the Author(s)

In this manuscript the authors introduce the use of foams derived from frogs as a drug delivery system for pharmaceutical applications. This is novel approach that could potentially be further developed. The authors have conducted a lot of experiemntal work including several analytical techniques. The discussion is well written and in a good agreement with the experimental data. I recommend the article for publication after some minor adjustments:

Response: We thank the referee for their positive comments on our work, recognising the novelty of our work and hopefully have addressed their concerns below.

-In Fig. 3 the authors should present the MTT findings as percentage (%) of the viable cells and not the absorbance values which are not easy to follow.

Response: We have replotted the data for the MTT as a percentage of the viable cells as suggested for Fig. 3 and updated the figure and legend in the manuscript. The raw data file submitted with the manuscript already had these data in and plotted so this does not need to be updated.

- What was the drug loading in the foams? This should be presented for the dyes and the drug substance used for the purpose of the work.

Response: The encapsulation efficacy was ~100%, as the drug-mimic or antibiotic used (400 μ l) was directly encapsulated/"mixed" with the foam at 1 mg/ml concentration. All free liquid was encapsulated by the foam following mixing. We have added additional text to clarify this in the Materials & Methods section (Lines 383-384, 402-403).

- How do the authors explain that the release patterns for the hydrophilic pigments is slower compared to the relative hydrophobic drugs rifampicin.

Response: We believe that the differences in release patterns are related to the this is due to the interaction between the drug molecule and the novel surfactant properties of these foam proteins. We have added some text (lines 224-233) to details this in light of previous structural work that was undertaken on one of the foam proteins. We believe that this is a further example of how unique this foam approach is for the delivery of both molecules.

Appendix C

Authors Response to comments on revised manuscript RSOS-210048.R1

Editor Comments to Author:

It is unusual to offer an additional opportunity to revise a paper in the journal, but the opportunity to revise has a number of caveats. Firstly, you should do your best to amend the paper to highlight the concerns raised by this reviewer, but the paper is probably publishable as long as it doesn't over claim. Given this, the second caveat is that you should change the title to "potential drug delivery system" or similar. Assuming you take these caveats into account, the paper should not need to go back to the critical reviewer, but will receive final assessment by the editors.

Response: We would like to thank the editor for giving us the opportunity to respond to the referees comments – We have revised the manuscript in response to the editor and referees comments, toning down the claims of a drug delivery system and further **emphasising the potential** of the frog foam to be used as pharmaceutical foam and possible use as drug delivery system and not over claiming (see various changes throughout the manuscript as tracked changes).

We have amended the title as suggested to appear as not to over claim also, suggesting that the foams share properties with pharmaceutical foams rather than being a 'drug delivery system'.

The title is now '**Frog nest foams exhibit pharmaceutical foam-like properties**'

We hope that the editor will now deem the work publishable.

Reviewer comments to Author:

Dear Editor,

I regret to say that the authors did not read my review correctly. The manuscript in its current, already "improved" form, does not represent state of the art work concerning rational development of medicines. It does not contain elements related to pharmaceutical technology. As mentioned in an earlier review:" Authors did not pay enough attention to the fact that the development of foams as unique drug delivery devices requires careful planning of all aspects including the selection of excipients, characterization techniques and the application.

Considering the above mentioned major drawbacks the manuscript in the present form as well as the generated data are not suitable for publication in the field of biopharmaceutics and pharmaceutical technology

Response: We thank the referee for the comments and we are sorry that we were unable to satisfy them in our previous responses. As we stated previously, we do not see this work as a finished pharmaceutical technology that is clinic ready (hence why we submitted to a journal of general interest rather than a Pharmaceutical Technology Journal), more that we have identified properties in a natural protein foam that are shared with pharmaceutical foams, and that they could find utility in drug delivery. We have carried out basic characterisation of the

properties of the foam and feel we have demonstrated that there is potential. Given that we don't envisage the wild-type foam being used as a delivery vehicle, our work attempts to demonstrate potential for use rather than describe a product, and we hope that may attract further work in the area. We have amended the manuscript further to try and convey this message.

There are still many elements that are irrational, e.g. why should the foam be stable for 168 hours? Why was rifampicin used as the model drug? It is an antibiotic reserved for the treatment of tuberculosis (not applied to the skin)? e.t.c.

Response: Whilst the it is difficult to respond to the 'many elements' mentioned by the referee that are not detailed, we have addressed the three specific points they have raised.

Line 268-269 – 'Why the foam should be stable for 168 hrs' – As can be seen in figure 5B, the release of calcein plateaus at around 85% of release, coinciding with 168 hrs (7 days). Given that most other stability measurements of pharmaceutical foams are in the order of minutes to hours, we didn't persist beyond 7 days, which is approaching the natural stable life of the foam. We have indicated this in the text and added references to support this finding (line 267-272).

Line 284-286 – we have added additional justification for the use of rifampicin as a model drug in the text. Antibiotics such as the polyketides clindamycin and tetracycline (ref 2 & 30) have previously been delivered using foam or gel preparations. Our model antibiotic, the polyketide rifampicin belongs to the same chemical class of antibiotics, but has the advantage of being pigmented and thus is spectroscopically amenable to assay – hence why we used it.

However, we would also like to point out that while rifampicin is used primarily for tuberculosis infections, it is not 'reserved for tuberculosis' and there are numerous and increasing uses of this antibiotic (often in combination) for treatment of *Staphylococcus* infections, as an anti-*Neisseria* prophylactic and for several other infections (for examples see <https://www.ncbi.nlm.nih.gov/pmc/articles/PMC2806656/> and references therein).